# The Roles and Regulation of m^6^A Modification in Glioblastoma Stem Cells and Tumorigenesis

**DOI:** 10.3390/biomedicines10050969

**Published:** 2022-04-22

**Authors:** Peng Li, Hope T. Richard, Kezhou Zhu, Linlin Li, Suyun Huang

**Affiliations:** 1Department of Human and Molecular Genetics, School of Medicine, Virginia Commonwealth University, Richmond, VA 23298, USA; peng.li@vcuhealth.org (P.L.); kezhou.zhu@vcuhealth.org (K.Z.); linlin.li@vcuhealth.org (L.L.); 2Department of Pathology, School of Medicine, Virginia Commonwealth University, Richmond, VA 23298, USA; hope.richard@vcuhealth.org; 3Institute of Molecular Medicine, School of Medicine, Virginia Commonwealth University, Richmond, VA 23298, USA; 4VCU Massey Cancer Center, Virginia Commonwealth University, Richmond, VA 23298, USA

**Keywords:** m^6^A modification, glioblastoma, epigenetics, methyltransferase, demethylase, inhibitor

## Abstract

Glioblastoma is the most common and most lethal primary malignant brain tumor. N^6^-methyladenosine (m^6^A) is a widespread and abundant internal messenger RNA (mRNA) modification found in eukaryotes. Accumulated evidence demonstrates that m^6^A modification is aberrantly activated in human cancers and is critical for tumorigenesis and metastasis. m^6^A modification is also strongly involved in key signaling pathways and is associated with prognosis in glioblastoma. Here, we briefly outline the functions of m^6^A and its regulatory proteins, including m^6^A writers, erasers, and readers of the fate of RNA. We also summarize the latest breakthroughs in this field, describe the underlying molecular mechanisms that contribute to the tumorigenesis and progression, and highlight the inhibitors targeting the factors in m^6^A modification in glioblastoma. Further studies focusing on the specific pathways of m^6^A modification could help identify biomarkers and therapeutic targets that might prevent and treat glioblastoma.

## 1. Introduction

Glioblastoma (CNS World Health Organization Grade 4) is a devastating type of primary brain tumor [1], with a median survival of only 14 months, regardless of treatment. Progress has been made in understanding glioblastoma, including many genetic and immunologic profiles described in recent years [2,3]; however, the molecular mechanisms and epigenetic alterations that regulate and drive glioblastoma development remain largely elusive.

N^6^-methyladenosine (m^6^A) modification, discovered and partially characterized in a great variety of cellular mRNAs in the 1970s [4], is the most essential and widespread internal methylation in eukaryotic mRNAs [5]. With the development and establishment of high-throughput sequencing technologies, such as methylated RNA immunoprecipitation sequencing (MeRIP-seq/m^6^A-seq) [6], m^6^A crosslinking immunoprecipitation sequencing (m^6^A-CLIP-seq) [7], m^6^A individual-nucleotide-resolution crosslinking, and immunoprecipitation (miCLIP) [8], the investigation of the m^6^A RNA methylomes and mapping are over 18000 m^6^A sites in the transcripts of more than 7000 human genes have been performed [6,9]. Methylation is a process of catalytically transferring a methyl group from an active methyl donor such as S-adenosylmethionine (SAM) to the substrate, which can chemically modify certain proteins or nucleic acids to form a methylated product [10]. The m^6^A modification has been proved to be reversible, as it involves methyltransferases (writer), demethylase (eraser), and m^6^A recognized RNA binding protein (reader). m^6^A regulates mRNA processing events such as alternative splicing, translation, and stability [11] (Figure 1). Increasing evidence shows that dysregulation of m^6^A modification and its corresponding proteins are implicated in the tumorigenesis and the progression of cancers, including glioblastoma [12,13] (Figure 2 and Table 1). These studies could provide potential epigenetic targets for the diagnosis and treatment of glioblastoma.

## 2. Writers in Glioblastoma

Installation of m^6^A is catalyzed by a methyltransferase complex (MTC) comprised of several proteins, known as m^6^A writers. The installation occurs at the pre-mRNA level during transcription and mRNA processing, and is completed when an mRNA is released from chromatin into the nucleoplasmic RNA [62]. The writers of m^6^A are mainly represented by a multi-subunit methyltransferase complex, which contains methyltransferase-like 3 (METTL3), methyltransferase-like 14 (METTL14), and Wilms tumor 1-associated protein (WTAP). Other components of the complex include RNA binding motif protein (RBM15/15B, VIRMA, synonym: KIAA1429), zinc finger CCCH-type containing 13 (ZC3H13), and Cbl proto-oncogene like 1 (CBLL1, synonym: HAKAI). METTL3 possesses methyltransferase activity, and METTL14 is catalytically inactive but serves as an RNA binding platform for METTL3. These two proteins are expressed together in the nucleus and form a stable heterodimer complex in the ratio of 1:1. Both METTL3 and METTL14 are required for m^6^A formation [63]. As the regulatory factor of m^6^A, WTAP facilitates the nuclear localization of METTL3 and METTL14. RBM15/RBM15B binds with METTL3 and participates in the regulation of m^6^A modification [64]. VIRMA is linked with alternative polyadenylation and mainly regulates m^6^A methylation of mRNA near the stop codon and 3′UTR [65]. ZC3H13, together with other cofactors, methylates three prime untranslated regions (3′UTR) [66]. CBLL1 interacts with the WTAP/ZC3H13/VIRMA complex and regulates some extents of m^6^A.

The expression of METTL3, significantly elevated in glioma stem-like cells (GSCs) and attenuated during differentiation, is the most important m^6^A writer [14]. Silencing METTL3 in GSCs inhibited neurosphere formation, enhanced sensitivity to γ-irradiation, and reduced DNA repair [14]. Mechanically, RNA immunoprecipitation (RIP) studies identified SRY-box transcription factor 2 (SOX2) as a bona fide substrate of METTL3 and the m^6^A modification of SOX2 mRNA enhanced its stability. Furthermore, METTL3 binding and m^6^A modification in vivo required intact METTL3/m^6^A sites present in the SOX2-3′UTR, resulting in the recruitment of human antigen R (HuR) to SOX2 mRNA to enhance its stability, GSCs maintenance, and dedifferentiation [14]. Interestingly, the latest research reported that METTL3 expression is positively associated with a higher grade and poorer prognosis of isocitrate dehydrogenase (IDH)-wildtype glioma but not IDH-mutant glioma. Furthermore, HuR was reported to be essential for METTL3-mediated stabilization of metastasis-associated lung adenocarcinoma transcript 1 (MALAT1), which subsequently activated nuclear factor kappa B (NFκB) in IDH-wildtype glioma [15].

Apart from directly regulating SOX2, an integrated approach of whole-transcriptome and m^6^A-RNA-immunoprecipitation-coupled sequencing (m^6^A-RIP Seq) of RNA isolated from GSCs with a non-targeting shRNA or METTL3 shRNA [16] found that there are novel RNA metabolic events that depend on METTL3 and the resultant m^6^A modification, which may work through the 3′UTR for regulation of RNA metabolism. METTL3 depletion dysregulates the expression of epigenetically activated genes, the RNA editing, particularly in exonic regions by altering the level of RNA editing enzymes (downregulating adenosine deaminase RNA specific (ADAR) and adenosine deaminase RNA specific B1 (ADARB1), while upregulating apolipoprotein B mRNA editing enzyme catalytic subunit 1 (APOBEC1) and apolipoprotein B mRNA editing enzyme catalytic subunit 3A (APOBEC3A)), and the splicing and stability in GSCs. Furthermore, the signaling pathways involved in GSCs maintenance and tumorigenesis, including Notch, NFκB, Wnt, c-Myc and TGF-β, are positively regulated by METTL3-mediated RNA stabilization, implying an oncogenic role for METTL3 in GSCs [16].

Additionally, it is reported that METTL3-mediated m^6^A modification was significantly elevated in temozolomide (TMZ)-resistant glioblastoma cells. Experimentally, METTL3 overexpression impaired the TMZ-sensitivity of glioblastoma cells, while METTL3 silencing or 3-deazaadenosine (DAA)-mediated total methylation inhibition improved the sensitivity of TMZ-resistant glioblastoma cells to TMZ in vitro and in vivo. This potentiation is likely because these two critical DNA repair genes (O-6-methylguanine-DNA methyltransferase (MGMT) and N-methylpurine DNA glycosylase (APNG)) were m^6^A-modified by METTL3 [17]. Another team illustrated that TMZ treatment induced upregulation of METTL3 and increased m^6^A modification. The underlying mechanism is that TMZ increased the chromatin accessibility at the METTL3 locus mediated by the EZH2-SOX4 complex [18].

Recently, it has been reported that the m^6^A levels in RNAs were lower in glioblastoma cells and tissues. Strikingly, the epithelial to mesenchymal transition (EMT) and vasculogenic mimicry (VM) process were enhanced after knockdown of METTL3, which was regulated by matrix metallopeptidase 2 (MMP2), cadherin 1, cadherin 2 and fibronectin 1 (FN1) [19].

Shi’s team revealed an increase in m^6^A RNA methylation upon induced GSCs differentiation. Meanwhile, knockdown of METTL3 or METTL14 promotes human GSCs growth, self-renewal, and tumorigenesis. In contrast, overexpression of METTL3 or inhibition of the RNA demethylase fat mass and obesity-associated protein (FTO) suppressed GSCs growth and self-renewal. Similarly, in vivo tumorigenesis experiments showed that knockdown of METTL3 and/or METTL14 significantly enhanced the growth of GSCs. In parallel, the authors found that meclofenaminc acid 2 (MA2), a selective inhibitor of FTO, increased the m^6^A level, diminished GSCs growth and self-renewal, and severely reduced GSCs-induced tumorigenesis. By mechanically combining m^6^A-seq analysis and RNA-seq in GSCs depletion of METTL3 or METTL14, they screened and verified a number of upregulated oncogenes, such as ADAM19, EPHA3, and KLF4, and downregulated tumor suppressors, including CDKN2A, BRCA2 and TP53I11. Additionally, the expression of differentiated neural cell markers (GFAP, Tuj1) was also decreased after knockdown of METTL3 or METTL14. The m^6^A consensus motif GGAC was identified and peak distribution analysis revealed strong enrichment of m^6^A peaks near the stop codon. Gene ontology (GO) analysis of genes with m^6^A peaks in their mRNA illustrated that m^6^A-methylated mRNAs are involved in critical cellular processing, such as cell growth, cell differentiation, DNA damage response, and cellular stress response [20]. Additionally, in 2021, Wang’s group found argininosuccinate synthase 1 (ASS1) was a target of METTL14-mediated m^6^A modification by MeRIP and RIP assay. Overexpression of METTL14 markedly elevated ASS1 mRNA m^6^A modification and suppressed ASS1 mRNA expression. Moreover, the authors revealed that METTL14-mediated ASS1 mRNA degradation relied on the YTH m^6^A RNA-binding protein 2 (YTHDF2)-dependant pathway [67]. 

Although originally classified as a tumor suppressor, WTAP was later found to be involved in the regulation of migration and invasion in glioblastoma [21]. Liu et al. reported that WTAP overexpression predicts poor prognosis in glioma [22], and as a crucial interactor of the methyltransferase complex it plays an oncogenic role in glioma.

## 3. Erasers in Glioblastoma

In 2011, FTO was discovered as the first m^6^A demethylase [68]. Then, another RNA demethylase, Alkb homolog 5 (ALKBH5), was identified in 2013 [69]. The findings of these studies showed that FTO and ALKBH5 belong to the alpha-ketoglutarate-dependent dioxygenase family. Bothe enzymes remove methyl groups from m^6^A in a Fe (II) and α-ketoglutaric acid-dependent mode. In the demethylation process, m^6^A is primarily oxidized to generate N^6^-hydroxymethyladenosine (hm^6^A), which is converted into N^6^-formyladenosine (f^6^A) subsequently. Finally, f^6^A is changed into adenosine (A), and the demethylation is completed.

Like the writers, m^6^A erasers also play key roles in glioblastoma. Knockdown of FTO in human glioma cell lines or via treatment with a selective inhibitor of FTO was thought to inhibit cell proliferation and migration [23]. Furthermore, FTO knockdown decreased the expression of MYC, increased the level of MAX interactor 1 (MXI1), and inhibited the primary and mature transcripts of miR-155-5p, miR-24-3p and miR-27a-3p, respectively. However, when human glioma cell lines were treated with 200 μmol/L MA2 for 2 days, a notable decrease in both mRNA and protein of MYC was observed, but there was no difference in FTO expression. In summary, these data indicated that FTO regulated the feedback loop in glioma cells by targeting MYC transcripts [24]. On the contrary, recent research is inconsistent with previous research. Li’s group reported that FTO expression in glioblastoma was lower than in low-grade glioma and normal brain tissue. Interestingly, overexpression of FTO restrained cell growth, migration and invasion in vitro and in vivo. The potential mechanism may be that FTO regulates the m^6^A modification of primary microRNA-10a (pri-miR-10a) processing [25].

As early as 2017, Huang’s group uncovered a critical function for ALKBH5 and provided insight into the important roles of m^6^A methylation in glioblastoma. The work showed that ALKBH5 is highly expressed in GSCs and in the patients’ samples compared to normal brain tissue. Interestingly, ALKBH5 co-expressed with the typical stemness marker for glioblastoma, SOX2 and Nestin in tumor tissue. Silencing ALKBH5 suppressed the proliferation of GSCs in vitro and tumorigenesis in vivo. Furthermore, combined transcriptome and m^6^A-seq data indicated that the transcription factor forkhead box M1 (FOXM1) was one of the ALKBH5 targeted genes. Specifically, ALKBH5 demethylates FOXM1 nascent transcripts, causing elevated FOXM1 expression. Meanwhile, a long non-coding RNA antisense to FOXM1 (FOXM1-AS) could promote the demethylation by enhancing the interaction between ALKBH5 and FOXM1 nascent transcripts [26].

In order to explore the m^6^A modification of glucose-6-phosphate dehydrogenase (G6PD) in the carcinogenesis of glioma, ALKBH5 was upregulated, which stimulated glioma cells to proliferate. ALKBH5 demethylated the target transcript G6PD and enhanced its mRNA stability, thereby promoting G6PD translation and activating the pentose phosphate pathway (PPP) [27].

In glioblastoma, rapid cancer cell growth and poorly organized tumor vasculature result in extensive hypoxia. In response to hypoxia, multiple subtypes of immunosuppressive cells and mainly tumor-associated microglia or macrophages (TAM) infiltrate the tumors in addition to the profound adaptation of cancer cells. Wu’s team reported that ALKBH5, along with the hypoxic gene, was induced in glioblastoma models under hypoxic conditions. Hypoxia-induced ALKBH5 removed m^6^A modification from the lncRNA NEAT1, resulting in a steady level of its transcript and an increasing paraspeckle assembly. This process gave rise to the relocation of transcriptional repressor splicing factor proline and glutamine-rich (SFPQ) from the C-X-C motif chemokine ligand 8 (CXCL8) promoter to paraspeckles, leading to the overexpression of CXCL8/ interleukin 8 (IL8). Suppression of ALKBH5 in glioblastoma cells significantly decreased CXCL8/IL8 production, diminished hypoxia-induced TAM recruitment, and immunosuppression in intracranial transplanted tumors. Collectively, the authors connected hypoxia-induced epitranscriptomic alters with the immunosuppressive microenvironment promoting tumor evasion [28].

The existence of GSCs resistance to radiotherapy and TMZ are the major causes of recurrence and poor prognosis of glioblastoma. One group demonstrates that high expression of ALKBH5 increased radioresistance by regulating homologous recombination (HR) in patient-derived GSCs. They observed a decrease in invasion and survival after irradiation likely due to a defect in DNA-damage repair in ALKBH5 knockdown cells, and a diminished expression of the genes involved in HR, such as CHK1 and RAD51, as well as the persistence of gamma H2A histone family member X (γ-H2AX) staining after insulation resistance (IR) [29]. Another group found that LncRNA SOX2OT was elevated in TMZ-resistant cells and recurrent glioblastoma patient samples, and this aberration was associated with a high risk of recurrence and poor prognosis. Depletion of SOX2OT inhibited cell proliferation and increased cell apoptosis thereby enhancing TMZ sensitivity. Mechanistically, ALKBH5 recruited by SOX2OT binds to SOX2 and demethylates the SOX2 transcript, leading to enhanced SOX2 expression and activating the Wnt5a/β-catenin signaling pathway [30]. Therefore, evidence is emerging that modulation of m^6^A RNA methylation plays an important role in the recurrence of glioblastoma.

## 4. Readers in Glioblastoma

### 4.1. YTHDF and YTHDC Families

Recognition and binding of m^6^A readers to their target RNAs is essential for modulating RNA fate. To date, the m^6^A readers discovered include YT521-B homology (YTH) domain-containing proteins, heterogeneous nuclear ribonucleoproteins (HNRNPS), insulin-like growth factor 2 mRNA binding protein (IGF2BPs) and eukaryotic initiation factor (eIF) 3 [70]. YTH family members include YTHDF1, YTHDF2, YTHDF3, YTHDC1 and YTHDC2. The YTH domain was contained at the C-terminus that harbors a binding site for m^6^A. YTHDF2 has been identified in the cytoplasm and nucleus, while other YTHDFs are generally located in the cytoplasm [71]. YTHDF1 and YTHDF3 promote translation in a cap-dependent manner via regulating the interaction of m^6^A at 3′-UTR with eukaryotic initiation factors (eIF3A and eIF3B) [72,73,74]. YTHDF2 selectively induces the degradation of m^6^A-modified mRNAs, lowering the abundance of its targets [75], while YTHDF3 enhances the decay and translation of RNAs by interacting with YTHDF1 and YTHDF2 [74,76]. YTHDC1 is a predominantly nuclear protein with a YTH domain and multiple other functional domains, including multiple putative nuclear localization elements and an SH2 domain. YTHDC1 appears to be the major reader of nuclear m^6^A to mediate RNA export and alternative splicing [77,78]. YTHDC2, a nucleocytoplasmic protein, contains a DEAD-box RNA helicase domain, raising translational activity and accelerating the degradation of its targets [79].

In 2020, Xu et al. confirmed that YTHDF1 is associated with glioma progression, and high expression of YTHDF1 predicts a poor prognosis in patients with glioma. This study identified *has-mir-346* as an upstream regulator of YTHDF1 to participate in the development of glioma [31]. Later, Yarmishyn et al. reported that knockdown of YTHDF1 in human glioblastoma cell lines inhibits proliferation, sensitizes glioblastoma cells to TMZ, and attenuates cancer stem cell-like properties. Additionally, Musashi-1 (MSI1) is a posttranscriptional gene expression regulator associated with high oncogenicity in glioblastoma that positively regulates the expression of YTHDF1 [32].

YTHDF2 was first reported in association with gliomas by informatics in 2020. At that time, high YTHDF2 expression was noted to be associated with poor overall survival in low-grade glioma (LGG) and positively correlated with several immune cells markers, including prephenate dehydratase 1 (PD-1), Hepatitis A virus cellular receptor 2 (HAVCR2/TIM-3), and cytotoxic T-lymphocyte associated protein 4 (CTLA-4), as well as TAM gene markers and IDH1 in these lesions [33]. Meanwhile, Huang’s group explored the mechanism responsible for the overexpression of YTHDF2 in glioblastoma. They discovered that epidermal growth factor receptor (EGFR) activation plays a role in m^6^A modification in glioblastoma and YTHDF2 overexpression occurred through the EGFR/SRC/ERK pathway. Phosphorylation of YTHDF2 serine39 and threonine381 by extracellular regulated MAP kinase (ERK) was found to stabilize the YTHDF2 protein, which is required for glioblastoma cell proliferation, invasion and tumorigenesis. Moreover, YTHDF2 facilitates m^6^A-dependent mRNA decay of L-xylulose reductase (LXRA) and HIVEP zinc finger 2 (HIVEP2), which impacts the glioma patient survival. Furthermore, YTHDF2 inhibits LXRα-dependent cholesterol homeostasis in glioblastoma cells [34]. Later, it was also confirmed that YTHDF2 accelerated UBX domain protein 1 (UBXN1) mRNA degradation via METTL3-mediated m^6^A, which, in turn, induced NFκB activation [35]. On the other hand, Rich’s group found that YTHDF2 stabilizes MYC and VEGFA transcripts in GSCs in an m^6^A-dependent manner, and IGFBP3 was identified as a downstream effector of the YTHDF2–MYC axis in GSCs. Thus, YTHDF2 links RNA epitranscriptomic modifications and GSCs growth, laying the foundation for the YTHDF2–MYC–IGFBP3 axis as a specific and novel therapeutic target in glioblastoma [36].

YTHDC1 also participates in the tumorigenesis of glioblastoma by reading the m^6^A marks dependent on its tryptophan 377 (W377) or W428 sites [80]. YTHDC1 deficiency inhibits sphere numbers substantially in METTL3 overexpressed U87 cells compared to the control. Overexpression of YTHDC1 W377A/W428A mutant also failed to enhance sphere formation of U87 cells, suggesting that YTFDC1 contributes to the glioblastoma phenotype dependent on its m^6^A-binding activity [38]. Finally, YTHDC1 inhibits cell proliferation by reducing the expression of Vacuolar protein-sorting-associated protein 25 (VPS25), which is highly expressed in gliomas and augments proliferation through targeting JAK-STAT signaling [39].

Although there are no reports documenting the function of YTHDF3 in glioblastoma, YTHDF3 is known to promote cancer cell interaction with brain endothelial cells and astrocyte blood–brain barrier extravasation, angiogenesis and outgrowth. Mechanistically, YTHDF3 enhances the translation of m^6^A-mediated transcripts for the genes associated with brain metastasis, such as ST6 N-acetylgalactosaminide alpha-2,6-sialyltransferase 5 (ST6GALNAC5), gap junction protein alpha 1 (GJA1), and EGFR. Overexpression of YTHDF3 in brain metastasis is attributed to the increased gene copy number and autoregulation of YTHDF3 cap-independent translation by binding to m^6^A residues within its own 5′UTR [37].

### 4.2. HNRNPs

In addition to the putative direct m^6^A readers in the YTH domain, there are indirect readers such as heterogeneous nuclear ribonucleoproteins (HNRNPs), which include HNRNPC, HNRNPG, and HNRNPA2/B1. The binding affinity of HNRNPC for transcripts can be increased by m^6^A-mediated RNA structure alteration, leading to the m^6^A-switch effect [81]. HNRNPA2/B1 was originally identified as a nuclear m^6^A-binding protein that modulates microRNA biogenesis. Although HNRNPA2/B1 lacks a YTH domain, the authors of this study suggested that it could be an m^6^A reader [82]. RNA-HNRNPG has been reported to be regulated by an abundance of m^6^A sites in the transcriptome, altering the expression and alternative splicing pattern of target mRNAs [83].

There are 144 proteins that have been identified using immunoprecipitation (IP)/MS which interact with SOX2 in glioblastoma cell lines. Among the proteins identified were HNRNPC and HNRPA2/B1. This finding indicates that SOX2 associates with a heterogeneous nuclear ribonucleoprotein complex [84]. Furthermore, HNRNPC was shown to be highly elevated in glioblastoma cell lines and brain tissue. Silencing of HNRNPC reduced cell proliferation and enhanced etoposide-induced apoptosis. Strikingly, silencing of HNRNPC lowered mi-R-21 levels in turn increased the expression of programmed cell death 4 (PDCD4), suppressing AKT serine/threonine kinase (Akt) and Ribosomal protein S6 kinase beta-1 (p70S6K) activation and inhibiting migration and invasion in glioblastoma [40]. Similarly, high expression of HNRNPC, verified in glioblastoma by Western blot, real-time polymerase chain reaction (RT-PCR) and immunohistochemical staining, was associated with malignancy and development of gliomas [85]. It was also confirmed that HNRPA2/B1 overexpressed in glioma tissue specimens is associated with advanced glioma grade and knockdown of HNRPA2/B1 leads to reduced glioblastoma cell viability, adhesion, migration, invasion, as well as chemoresistance for TMZ. HNRPA2/B1 knockdown also induced apoptosis and reactive oxygen species (ROS) generation in glioma U251 and SHG44 cells. Molecularly, phospho-STAT3, MMP2, and matrix metallopeptidase 9 (MMP9) are the targets of HNRPA2/B1 in glioma [41,42]. In addition, human cytomegalovirus (HCMV) immediate early 86 protein (IE86, ie2 gene-encoded) promotes glioblastoma migration by regulating HNRPA2/B1 expression [86].

### 4.3. IGF2BP Family

Insulin growth factor 2 mRNA binding proteins (IGF2BPs), including IGF2BP1/2/3, also function as indirect m^6^A readers to inhibit the degradation of m^6^A-modified transcripts and facilitate translation [87].

IGF2BP1 is upregulated in human glioma tissue, which is associated with cell proliferation, migration, invasion, and cancer progression. Some non-coding RNAs, including miR-4500 [43], miR-837 [44], miR-506 [45], miR-513 [46], Lnc-THOR [47], and LINC00689 [48], promote tumorigenesis in glioma through targeting IGF2BP1.

IGF2BP2 is also overexpressed in glioblastoma. It regulates oxidative phosphorylation (OXPHOS) in primary glioblastoma sphere cultures. IGF2BP2 binds several mRNAs that encode mitochondrial respiratory chain complex subunits and interact with complex I proteins. Furthermore, depletion of IGF2BP2 in glioma spheres decreases their oxygen consumption rate through compromising complex I and complex IV activity, which results in impaired clonogenicity in vitro and tumorigenicity in vivo [49]. Additionally, the long non-coding RNA, HIF1A-AS2, OIP5-AS1, and CASC9 target IGF2BP2, both directly and indirectly, to facilitate the maintenance of mesenchymal glioblastoma stem-like cells [50], to repress resistance to TMZ and proliferation via OIP5-AS1-miR-129-5p axis [51], and to accelerate aerobic glycolysis by enhancing Hexokinase-2 (HK2) mRNA stability [52]. Recently, it has been reported that serine/arginine-rich splicing factor 7 (SRSF7), an oncogenic splicing factor, promotes the proliferation and migration of glioblastoma cells dependent on the presence of the m^6^A methyltransferase. Furthermore, the two m^6^A sites of PDZ binding kinase (PBK) are regulated by SRSF7 in glioblastoma cells through recognition by IGF2BP2 [53]. IGF2BP2 also inhibits the phosphotyrosine interaction domain containing 1 (PID1) expression through the DANCR/FOXO1 axis, leading to drug resistance in glioblastoma cells and promoting glioma progression [54].

The mRNA and protein expression of IGF2BP3 is up-regulated in glioblastomas but not in lower-grade astrocytomas. IGF2BP3 is an RNA binding protein known to bind to the 5′UTR of insulin-like growth factor (IGF-2) mRNA, activating its translation without altering its transcript level. Gain and loss of function studies in glioblastoma cells established the role of IGF2BP3 in promoting proliferation, anchorage-independent growth, invasion, and chemoresistance. Concordantly, PI3K and MAPK, the downstream effectors of IGF-2, are activated by IGF2BP3 and essential for IGF2BP3-induced cell proliferation [55]. Functional assay confirmed IGF2BP3 as a target of miR-129-1, while targeting of IGF2BP3 by miR-129-1 decreases the MAPK/ERK and PI3K/AKT cascade which leads to the postponement of G1-S transition in the cell cycle in glioblastoma cells [56]. Other studies illustrated that miR-654 was identified as a target of circHIPK3 while IGF2BP3 was targeted by miR-654. CircHIPK3 promoted IGF2BP3 expression via interacting with miR-654 in glioma cells [57]. IGF3BP2 participates in a positive feedback loop of the lncRNA-RMRP/ZNRF3 axis and Wnt/beta-catenin signaling to regulate progression and TMZ resistance in gliomas. It is also involved in macrophage infiltration into the tumor microenvironment (TME) via stabilizing circNEIL3 packaged into exosomes by hnRNPA2B1 [58].

### 4.4. eIF3 Family

Eukaryotic translation initiation factor eIF3, a multimeric complex composed of 13 subunits (a-m), connects the 43S pre-initiation and eIF4F initiation complexes. Novel insights into the link between eIF3 and cancer have shown that eIF3 specifically recognizes mRNA structures, mRNA modifications, or the 5′cap of mRNAs [88]. A study of eIF3B in glioblastoma showed knockdown of eIF3 significantly inhibited proliferation in U87 cells, which was associated with the accumulation of G0/G1-phase cell numbers and an increased rate of apoptosis [59]. Similar to the function of eIF3B, eIF3D promotes cell growth, colony formation and migration in U251 and U87 cells [60]. Additional studies revealed that the mRNA of eIF3B, eIF3I, eIF4A1, eIF4E, and eIF4H was significantly higher in gliomas compared to non-neoplastic cortical control brain tissue (CCBT), but only eIF3I and eIF4H were significantly associated with patient outcome according to the cancer genome atlas (TCGA) dataset [60,61]. eIF3E has been found to be essential for the proliferation and survival of glioblastoma cells through the regulation of hypoxia-inducible factors (HIFs) [89].

## 5. Potential Clinical Inhibitors of m^6^A Modification in Glioblastoma

The above description indicates that the writers, erasers, and readers in m^6^A modification are critically important for tumorigenesis and aggressiveness by regulating structure stability, RNA processing, translation, degradation, and nuclear export in glioblastoma. A novel perspective for individualized therapy of glioblastoma and other tumors may include targeting the regulators of m^6^A modification.

The functional domain of METTL3 that binds to the substrate of METTL3 was thought to be the target of designed inhibitors. Recently, METTL3 inhibitors, the competitive inhibitors binding the pocket of SAM, can be divided into two types: nucleosides (compounds) and non-nucleosides (UZH1A, UZH2 and STM2457). UZH1a was able to reduce m^6^A/A level at least 6 days into an mRNA fraction in a leukemia cell line but also in other cell lines (U2OS, HEK293T) [90]. The effect of METTL3 inhibitors has not been reported on glioblastoma, but we believe that it is worthy of future study.

FTO inhibitors, FTO-04, FTO-10, FTO-11, and FTO-12, reduced the size of neurospheres treated at a concentration of 30 μM. Moreover, FTO-04 can significantly impair the self-renewal of GSCs to prevent neurosphere formation without impairing the growth of human neural stem cell (hNSC) neurospheres. Importantly, treatment with FTO-04 was found to increase the levels of both m^6^A and m^6^Am modifications, with m^6^Am modifications showing the largest fold-change relative to DMSO control. These findings make FTO-04 a leading compound for future therapy in glioblastoma [91].

MV1035, an ALKBH5 inhibitor, can reduce U87-MG migration and invasiveness via down-regulation of CD73, which is an extrinsic protein involved in the generation of adenosine and overexpressed in glioblastoma [92]. Under the computer-aided development of active inhibitors of ALKBH5, Simona et al. identified two low micromolar compounds among the in silico-predicted compounds by using an m^6^A antibody-based enzymatic assay. These two compounds, 2-[(1-hydroxy-2-oxo-2-phenylethyl) sulfanyl] acetic acid and 4-[(furan-2-yl)-methyl] a-1,2-diazinane-3,6-dione, reduced cell viability from 100% down to about 40% at low micromolar concentrations [93].

Lastly, a small molecule inhibitor termed 7773 interacts with a hydrophobic surface at the boundary of IGF2BP1 KH3 and KH4 domains and inhibits binding to Kras mRNA, causing a reduction in Kras mRNA and other RNA targets. This decreases cell migration and growth in cancer cell lines (H1299, ES2, and HEK293) [94]. Kras is also upregulated in glioma samples and is involved in the ERK pathway in gliomas [95]. Therefore, IGF2BP1 inhibitors may also be promising small molecule inhibitors for glioma treatment in the future.

## 6. Conclusions

Increasing evidence reveals that the writers, erasers, and readers of m^6^A modification play an important role in the development and tumorigenesis of glioblastoma. Moreover, some of the writers, erasers, and readers in m^6^A modification are potential biomarkers for diagnosis and promising drug targets for therapy in these lesions. Because glioblastoma is proven to be driven by epigenetic alteration, including mRNA manipulating the expression of oncogenes, targeting m^6^A regulatory proteins can serve as a new approach for precisely modifying the epitranscriptome of glioblastoma and lead to a more personalized approach to glioblastoma treatment. With the development of high-throughput sequencing technologies and chemical genetics, the studies of epitranscriptome on m^6^A modification have shed light on the therapy of glioblastoma. The development of inhibitors of the factors in m^6^A modification can control the deposition and removal of m^6^A marks which control RNA fate and specify the RNA to be regulated. Future efforts should be directed to the development of new inhibitors targeting the above m^6^A modification factors and validating their effects on glioblastoma treatment.

## Figures and Tables

**Figure 1 biomedicines-10-00969-f001:**
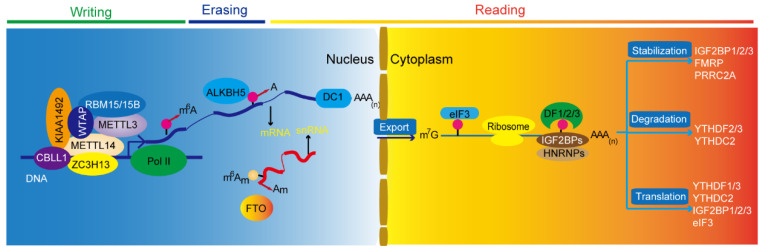
m^6^A modification determines RNA life fate. Here, is the cycling model of methyltransferase complex, demethylase, and m^6^A binding proteins. A, adenosine; m^6^A, N^6^-methyladenosine; m^6^Am, N^6^, 2′-O-dimethyladenosine; Am, 2′-O-methyladenosine; snRNA, small nuclear RNA; m^7^G, N^7^-methylguanosine; AAA (n), polyadenylation.

**Figure 2 biomedicines-10-00969-f002:**
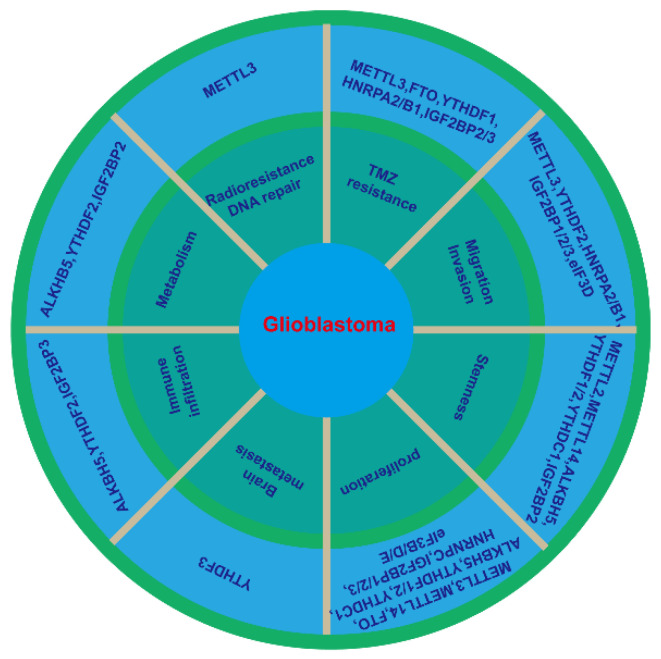
m^6^A regulator and the hallmarks of glioblastoma. Abnormal expression of m6A writers, erasers and readers in glioblastoma affects at least eight of the hallmarks: cell proliferation, stemness, migration and invasion, TMZ resistance, radioresistance and DNA repair, metabolism, immune infiltration, and brain metastasis.

**Table 1 biomedicines-10-00969-t001:** List of reported functions of m^6^A regulatory proteins in glioblastoma.

Gene Name	Role in RNA Modification	Role in Glioblastoma	Mechanism	References
METTL3	writer	oncogene	inhibiting sensitivity to γ-irradiation and enhancing DNA repair through recruitment of HuR to SOX2 mRNA.	[14]
		oncogene	Activating NFκB in IDH-wildtype glioma after stabilization of MALAT1	[15]
		oncogene	Dysregulating the expression of epigenetically activated genes (the RNA editing, spicing and stability)	[16]
		oncogene	Impairing the TMZ-sensitivity through m^6^A-modified DNA repair genes (MGMT and APNG)/EZH2	[17,18]
		Suppressor	Inhibiting epithelial to mesenchymal transition (EMT) and vasculogenic mimicry	[19]
		Suppressor	promoting cell growth, cell differentiation, DNA damage response and cellular stress response by enhancing m^6^A	[20]
METTL14	writer	Suppressor	promoting cell growth, cell differentiation, DNA damage response and cellular stress response by enhancing m^6^A	[20]
WTAP	writer	oncogene	A crucial interactor of the methyltransferase complex	[21,22]
FTO	eraser	oncogene	promoting proliferation and migration	[23]
		oncogene	Increasing cell proliferation by targeting MYC transcripts	[24]
		Suppressor	Inhibiting cell growth, migration and invasion by regulating m6A modification of primary pri-miR-10a processing	[25]
ALKBH5	eraser	oncogene	Inhibiting cell proliferation and stemness through demethylating FOXM1 nascent transcripts and increasing HuR binding	[26]
		oncogene	Demethylating G6PD transcript and enhancing its mRNA stability	[27]
		oncogene	Enhancing hypoxia-induced TAM recruitment and immunosuppression by CXCL8/IL8	[28]
		oncogene	Increasing radioresistance by regulation homologous recombination	[29]
		oncogene	Regulating TMZ resistance by promoting SOX2 expression	[30]
YTHDF1	reader	oncogene	Promoting cell proliferation, stemness, and TMZ resistance via Musashi-1	[31,32]
YTHDF2	reader	oncogene	Positively correlating with immune cells markers, TAM markers and IDH1	[33]
		oncogene	Inhibiting cell proliferation, invasion and tumorigenesis through EGFR/SRC/ERK	[34]
		oncogene	Accelerating UBXN1 mRNA degradation via METTL3-mediated m^6^A	[35]
		oncogene	Linking epitranscriptomic modification by stabilizing MYC and VEGFA transcripts	[36]
YTHDF3		oncogene	Promoting brain metastasis through enhancing the translation of m^6^A-mediated transcripts (ST6GALNAC5, GJA1 and EGFR)	[37]
YTHDC1	reader	oncogene	Promoting cell proliferation and stemness through VPS25-JAN-STAT	[38,39]
HNRNPC	reader	oncogene	Promoting cell proliferation, migration and invasion, and inhibiting apoptosis through Akt and p70S6K activation.	[40]
HNRPA2/B1	reader	oncogene	Increasing cell viability, adhesion, migration, invasion, and TMZ resistance, and inhibiting apoptosis and ROS targeting STAT3, MMP-2/9	[41,42]
IGF2BP1	reader	oncogene	Targeted by non-coding RNAs and promoting cell proliferation, migration, and invasion	[43,44,45,46,47,48]
IGF2BP2	reader	oncogene	Maintaining stemness and cell proliferation by regulating OXPHOS	[49]
		oncogene	Targeted by non-coding RNAs and increasing TMZ resistance and proliferation	[50,51]
		oncogene	Accelerating aerobic glycolysis by enhancing HK2 mRNA stability	[52]
		oncogene	Promoting proliferation, and migration through recognition SRSF7	[53]
		oncogene	Promoting drug resitance by inhibition of PID1 through DANCR/FOCO1 axis	[54]
IGF2BP3	reader	oncogene	Promoting proliferation, invasion and chemoresistance through PI3K and MAPK activation	[55]
		oncogene	Targeted by miR-129-1 and miR-654 to induce proliferation and TMZ resistance	[56,57]
		oncogene	Involving in macrophage infiltration in TME via stabilizing circNEIL3	[58]
eIF3B	reader	oncogene	Promoting proliferation and inhibiting apoptosis	[59]
eIF3D	reader	oncogene	Promoting cell growth, colony formation and migration	[60]
eIF3E	reader	oncogene	Promoting proliferation through HIFs	[61]

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
