# Peer review of "The Roles and Regulation of m6A Modification in Glioblastoma Stem Cells and Tumorigenesis"

_biomedicines, 2022, doi:10.3390/biomedicines10050969_

Round 1

Reviewer 1 Report

The review "The Roles and Regulation of m6A Modification in Glioblastoma Stem Cells and Tumorigenesis" by Li et al. is aimed to provide a comprehensive overview of the published data regarding  the role of epitranscriptome in glioblastoma. The most abundant mRNA modification, m6A, plays an important role in glioblastoma progression by regulating specific mRNAs. The involvement of writers, erasers and readers in glioblastoma progression has been reviewed, as well as putative applicability of their modification in developing novel treatment modalities for glioblastoma. However, parts of this manuscript are difficult to follow and need to be carefully corrected to avoid the occurrence of incomplete sentences (e.g. " While, RBM15/RBM15B interacts with METTL3 in a 77 WTAP-independent manner and is involved in the regulation of m6A modification of 78 some RNAs [16]."). For easier reading , the abbreviations  should be defined when first mentioned in the text. 

Author Response

We thank the assistant editor Kaylin Long and reviewers for thorough evaluation of our review. We found all the comments and suggestions to be insightful and constructive. We have followed Kaylin Long and the revierers’ suggestions to revise the manuscript. We believe that we are able to address all the critiques adequately.

The point-by-point responses are provided as following.

Reviewer 1

The review "The Roles and Regulation of m6A Modification in Glioblastoma Stem Cells and Tumorigenesis" by Li et al. is aimed to provide a comprehensive overview of the published data regarding the role of epitranscriptome in glioblastoma. The most abundant mRNA modification, m6A, plays an important role in glioblastoma progression by regulating specific mRNAs. The involvement of writers, erasers and readers in glioblastoma progression has been reviewed, as well as putative applicability of their modification in developing novel treatment modalities for glioblastoma.

Response: We thank the reviewer for recognizing the importance of our review.

However, parts of this manuscript are difficult to follow and need to be carefully corrected to avoid the occurrence of incomplete sentences (e.g. " While, RBM15/RBM15B interacts with METTL3 in a 77 WTAP-independent manner and is involved in the regulation of m6A modification of 78 some RNAs [16]."). For easier reading, the abbreviations should be defined when first mentioned in the text.

Response:

We thank the reviewer for the suggestions.

We checked all the sentences thoroughly and made the required modifications.

For instance:

  1. “While, RBM15/RBM15B interacts with METTL3 in a WTAP-independent manner and is involved in the regulation of m6A modification of some RNAs [16].” was changed into “Whereas, RBM15/RBM15B binds with METTL3 and participates in the regulation of m6A modification [16]”.
  2. All the abbreviation were defined and added in the part of Abbreviation.
  3. Other modifications can be tracked in tracking version of manuscript.

Reviewer 2 Report

In this paper Peng Li and colleagues reviewed the wealth of knowledge regarding the role of m6A modification, its regulation through several proteins (such as RNA writers, erasers, and readers) and its contribution in glioblastoma (GBM) tumorigenesis. Overall, the paper is nicely written, informative, and paves the next steps to identify novel biomarkers and therapeutic targets in GBM biology.

However, some parts of the paper contain very long and verbose sentences. The authors should re-phrase some parts by lightening or breaking up the text; this would make reading smoother and easier for the reader to follow. Moreover, in several parts the related references are missing. The reference should not only be reported in the table, but also in the text when appropriate.

Line 112: please add the related references; Lines 115-127: please add the missing references; Lines 153-159: please add references; Line 179-190: please add references; and so on.

Finally, minor issues: there are some places with grammatical errors/spelling errors. Some examples are reported below. There are some minor issues, which are reported below, that need to be addressed or fixed.

Abstract:

- Line 43: “…can chemically modified..”; please correct into “modify”

Paragraph 2:

- Line 65: writers;

Line 75: form

Line 91, 108: reported that..

Line 108: “…modification were significantly..”; please correct; modification was or modifications were.

Line 109: TMZ should be spelled out.

Paragraph 3:

- Line 146: “..are belong”; please correct into belong

- Line 265; highly

- Line 299: phosphor-STAT3; please correct the typo

Author Response

We thank the assistant editor Kaylin Long and reviewers for thorough evaluation of our review. We found all the comments and suggestions to be insightful and constructive. We have followed Kaylin Long and the revierers’ suggestions to revise the manuscript. We believe that we are able to address all the critiques adequately.

The point-by-point responses are provided as following.

Review 2

In this paper Peng Li and colleagues reviewed the wealth of knowledge regarding the role of m6A modification, its regulation through several proteins (such as RNA writers, erasers, and readers) and its contribution in glioblastoma (GBM) tumorigenesis. Overall, the paper is nicely written, informative, and paves the next steps to identify novel biomarkers and therapeutic targets in GBM biology.

Response:

We thank the reviewer for the positive comments.

However, some parts of the paper contain very long and verbose sentences. The authors should re-phrase some parts by lightening or breaking up the text; this would make reading smoother and easier for the reader to follow. Moreover, in several parts the related references are missing. The reference should not only be reported in the table, but also in the text when appropriate.

Line 112: please add the related references; Lines 115-127: please add the missing references; Lines 153-159: please add references; Line 179-190: please add references; and so on.

Response:

We thank the review for pointing out the missing references and valuable suggestions.

  1. Reference 23 (Li, F.X., et al., Interplay of m(6)A and histone modifications contributes to temozolomide resistance in glioblastoma. Clinical and Translational Medicine, 2021. 11(9).) was added to illustrate TMZ increased the chromatin accessibility at METTL3 locus mediated by EZH2-SOX4 complex.
  2. Reference 24 (Tao, M., et al., Decreased RNA m(6)A methylation enhances the process of the epithelial mesenchymal transition and vasculogenic mimicry in glioblastoma. American Journal of Cancer Research, 2022. 12(2): p. 893-+.) was added to report the association between METTL3 and epithelial to mesenchymal transition (EMT) and vasculogenic mimicry (VM) process.
  3. Reference 32 (Li, F., C. Zhang, and G. Zhang, m6A RNA Methylation Controls Proliferation of Human Glioma Cells by Influencing Cell Apoptosis. Cytogenetic and Genome Research, 2019. 159(3): p. 119-125.) and 34 (Zhang, S., et al., SPI1-induced downregulation of FTO promotes GBM progression by regulating pri-miR-10a processing in an m6A-dependent manner. Molecular Therapy - Nucleic Acids, 2022. 27(2162-2531 (Print)): p. 699-717.) were added to reveal the writer FTO has different functions in glioma.

Finally, minor issues: there are some places with grammatical errors/spelling errors. Some examples are reported below. There are some minor issues, which are reported below, that need to be addressed or fixed.

Abstract:

- Line 43: “…can chemically modified...”; please correct into “modify”

Response: Following the reviewer suggestion. “modified” was corrected into “modify”.

Paragraph 2:

- Line 65: writers;

Line 75: form

Line 91, 108: reported that…

Line 108: “…modification were significantly...”; please correct; modification was or modifications were.

Line 109: TMZ should be spelled out.

Response: We thank reviewer for careful corrections.

Line 65: “writer” was replaced by “writers”;

Line 75: “form” was corrected with “forms”;

Line 91, 108: “it’s reported …” was changed into “it’s reported that…”;

Line 108: “…modification were significantly…” was changed into “…modification was significantly…”;

Line 109: “TMZ” was spelt out “temozolomide”.

Paragraph 3:

- Line 146: “..are belong”; please correct into belong

- Line 265; highly

- Line 299: phosphor-STAT3; please correct the typo

Response: We thank the reviewer for the suggestions.

Line 146: “...are belong” was corrected into “belong”;

Line 265: “high” was replaced with “highly”;

Line 299: “phosphor-STAT3” was changed into “phospho-STAT3”.

Round 2

Reviewer 1 Report

The authors improved the manuscript according to reviewers' recommendations.